# A Winter-to-Summer Transition of Bacterial and Archaeal Communities in Arctic Sea Ice

**DOI:** 10.3390/microorganisms10081618

**Published:** 2022-08-10

**Authors:** Stefan Thiele, Julia E. Storesund, Mar Fernández-Méndez, Philipp Assmy, Lise Øvreås

**Affiliations:** 1Department of Biological Science, University of Bergen, Thormøhlensgate 53 A/B, 5020 Bergen, Norway; 2Bjerknes Centre for Climate Research, Jahnebakken 5, 5007 Bergen, Norway; 3Institute of Marine Research, Nordnesgaten 50, 5005 Bergen, Norway; 4Norwegian Polar Institute, Fram Centre, Hjalmar Johansens Gate 14, 9296 Tromsø, Norway; 5Biological Oceanography, GEOMAR Helmholtz Centre of Ocean Research, Düsternbrooker Weg 20, 24105 Kiel, Germany; 6Department of Arctic Biology, University Center in Svalbard, UNIS, 9171 Longyearbyen, Norway

**Keywords:** Arctic sea ice, biodiversity, microbial ecology, arctic microbes, N-ICE2015, Nitrosopumilus, sea-ice algal bloom

## Abstract

The Arctic is warming 2–3 times faster than the global average, leading to a decrease in Arctic sea ice extent, thickness, and associated changes in sea ice structure. These changes impact sea ice habitat properties and the ice-associated ecosystems. Sea-ice algal blooms provide various algal-derived carbon sources for the bacterial and archaeal communities within the sea ice. Here, we detail the transition of these communities from winter through spring to early summer during the Norwegian young sea ICE (N-ICE2015) expedition. The winter community was dominated by the archaeon *Candidatus* Nitrosopumilus and bacteria belonging to the *Gammaproteobacteria* (*Colwellia*, *Kangiellaceae,* and *Nitrinocolaceae*), indicating that nitrogen-based metabolisms, particularly ammonia oxidation to nitrite by *Cand.* Nitrosopumilus was prevalent. At the onset of the vernal sea-ice algae bloom, the community shifted to the dominance of *Gammaproteobacteria* (*Kangiellaceae, Nitrinocolaceae*) and *Bacteroidia* (*Polaribacter*), while *Cand.* Nitrosopumilus almost disappeared. The bioinformatically predicted carbohydrate-active enzymes increased during spring and summer, indicating that sea-ice algae-derived carbon sources are a strong driver of bacterial and archaeal community succession in Arctic sea ice during the change of seasons. This implies a succession from a nitrogen metabolism-based winter community to an algal-derived carbon metabolism-based spring/ summer community.

## 1. Introduction

The increase in global average temperature has pronounced effects on the Arctic [1], leading to a decline in sea-ice extent and thickness and to a change in the sea-ice structure in the Arctic Ocean. The change in ice thickness, quality, and decline in extent has potentially dramatic effects on various aspects of the Arctic ecosystem [2,3,4]. In 2015, the Norwegian young sea ICE (N-ICE2015) expedition investigated the Arctic sea ice during five and a half months, spanning from Arctic winter to spring and early summer, to investigate the status quo of the atmosphere, the ice, the ocean, and the interconnected ecosystems in the Arctic [5].

During the N-ICE expedition, a pelagicphytoplankton bloom was observed under the ice from late May to late June [6]. This bloom was dominated by the haptophyte *Phaeocystis pouchetii* [6]. The bacterial and archaeal community composition in the water column was investigated, covering the transition from winter to early summer, including the phytoplankton bloom [7]. The bacterial community consisted mainly of *Alpha*- and *Gammaproteobacteria* and changed from a dominance of *Candidatus* Pelagibacter during winter to the dominance of *Gammaproteobacteria* in spring and early summer [7]. This change in community composition can be attributed to increased carbon source availability derived from the phytoplankton bloom, as such changes have been monitored previously [7,8]. The most prominent difference to temperate waters was found in the high abundance of *Candidatus* Nitrosopumilus, a member of the *Nitrososphaeria* (formerly classified as both Marine Group I *Crenarchaeota* and Marine Group I *Thaumarchaeota*), in the surface samples in winter and a near absence in the summer samples. This decrease is most likely due to photo-inhibition of members of this group at high light regimes in spring and summer and competition with phytoplankton for ammonium in the surface ocean [7,9,10].

In addition to the under-ice phytoplankton bloom, the sea-ice algae bloom dominated by *Nitzschia frigida* was found at the bottom of second-year ice (SYI) [11]. Such blooms are, similarly to pelagic blooms, replenishing the carbon reservoir and enriching the sea-ice environment with easily degradable carbon sources. These carbon sources provide nutrients and energy sources for different members of the bacterial and archaeal community [7,8]. 

Despite the low temperatures and high salinity in the brine channels, bacterial and archaeal communities thrive in these environments, both in Arctic and Antarctic sea ice. The communities are mostly dominated by *Flavobacteriia* and *Gammaproteobacteria*, but also *Alphaproteobacteria*, *Verrucomicrobia*, and *Bacilli* [12,13,14,15,16]. In winter, a high abundance of *Crenarchaea* of the Marine Group I was found in ice and the underlying water, suggesting an abundance of the ammonia-oxidizing *Candidatus* Nitrosopumilus, which have been classified in this group previously [17,18]. This would suggest more chemolithotrophic metabolisms in winter. In spring and summer metabolisms based on algal-derived carbon sources seem likely. Bacterial and archaeal communities, particularly members of the *Gammaproteobacteria* (especially *Glaciecola* and *Colwellia*) and *Flavobacteriia* (especially *Polaribacter*), have been found in high abundance in connection to phytoplankton blooms in marine environments and similar blooms of sea-ice algae, e.g., *Fragilariopsis cylindrus*, *Nitzschia frigida*, or *Melosira arctica*, which occur in the bottom layers of sea ice in spring/summer [7,8,18,19,20]. Generally, *Bacteroidia* and *Gammaproteobacteria* have been found in high abundance in Arctic sea ice and brine samples [16,21,22]. However, most samples were analyzed at a higher taxonomic resolution; samples were taken during the summer month and not in winter, and at different geographic locations, including mostly coastal environments [16,21,22]. This might significantly alter the sea-ice bacterial and archaeal communities. 

The transition of bacterial and archaeal communities in sea ice from winter to summer is widely unknown. Such studies could elaborate on the status of algal-fueled microbial ecosystems and potentially enable conclusions about the changes in these ecosystems regarding the ongoing effects of global warming. Here, we present a seasonal investigation of microbes in Arctic sea ice from winter to spring and early summer of 2015, hypothesizing an algal-derived carbon sources triggered the succession of the bacterial and archaeal communities.

## 2. Materials and Methods

### 2.1. Ice Core Sampling

Ice cores were sampled during the Norwegian young sea ICE (N-ICE2015) expedition from 15 January to 22 June 2015 in the Arctic sea ice pack north of Svalbard, Norway, on board *RV Lance* [5]. During the drift, in total, 7 ice cores were taken from SYI on Floes 1–3 and FYI on Floe 4: 29 January (Floe 1; 83.063° N, 17.582° E), 5 March (Floe 2; 83.142° N, 24.1277° E), 12 March (Floe 2; 82.9296° N; 21.4409° E), 22 April (Floe 3; 82.8478° N, 16.5763° E), 21 May (Floe 3; 81.2328° N, 9.6369° E), 4 June (Floe 3; 80.2939° N, 4.0200° E), and 17 June (Floe 4; 80.4759° N, 7.8683° E; Figure 1). All samples from 29 January to 22 April (Floes 1, 2, and 3) were classified as winter samples, the sample from 21 May (Floe 3) was classified as spring sample, and the samples on 4 and 17 June (Floe 3 and 4) as early summer samples [23]. Due to the necessary changes of Floes, the displayed data do not represent a true time series sampled from a single ice Floe but rather four different ice Floes. The samples were taken using a 14 cm Mark II coring system (KOVACS enterprise, Roseburg, OR, USA). The bottom 0–10 cm of the core were cut, transferred into a sterile plastic bag, and stored at −20 °C until further processing. The core parts were thawed for 18–24 h in the dark at 4 °C, resulting in ~1.2–1.3 L of water. Even though this may lead to a shift in the community due to the long melting time, this is not a major issue since bacterial and archaeal growth at this temperature is very low and should not enable fast community changes. In addition, an osmotic shock due to the melting cannot be excluded but seems unlikely given the resulting marine community structure. A total of 1.8 mL of water was set aside for flow cytometry (FCM) analyses, and the remaining water was filtered through 0.2 µm pore size Sterivex filters (Merck, Oslo, Norway). The filters were immediately frozen in liquid nitrogen and then stored at −80 °C. Nitrate and Chl a data were retrieved from the general dataset by Assmy [24].

### 2.2. Bacterial Cell Numbers

The triplicates of 1.8 mL FCM samples were fixed with glutaraldehyde (0.5% final conc.) at 4 °C for a minimum of 2 h and then flash frozen in liquid nitrogen and stored at −80 °C until later analyses. Abundance of bacteria was determined on an Attune Acoustic Flow cytometer (Applied Biosystems by Life technologies, Waltham, MA, USA) with a syringe-based fluidic system using a 20 mW 488 nm blue laser after SYBR-Green staining. Prior to analyses, the samples were thawed and stained with SYBR Green I (Molecular Probes, Eugene, OR, USA) for a minimum of 1 h. Heterotrophic bacteria and archaea were analyzed using a flow rate of 25 µL min^−1^ and HNF at 500 µL min^−1^, following protocols modified from Marie et al. [25]. 

### 2.3. DNA and RNA Extraction and Sequencing

For phylogenetic analyses of the bacterial and archaeal diversity and a corresponding inference of their activity, the 16S rRNA genes were sequenced from DNA and RNA extracted from the samples. For that, the Sterivex filters were thawed on ice, and subsequently, a lysis solution was added. DNA and RNA were extracted from filters using the AllPrep DNA/RNA Mini Kit (Qiagen, CA, USA) with a modified protocol as described in Wilson et al. [26]. Briefly, 1 mL extraction buffer was added to each filter, and the filters were vortexed vertically for 2 min, inverted, and vortexed for 2 min. The lysate was removed from the filter and transferred to a 1.5 mL microcentrifuge tube. An AllPrep DNA spin column was loaded with 700 µL lysate and centrifuged for 30 s at 8000× *g*, saving the flow-through for subsequent RNA extraction. Centrifugation steps were repeated for the remaining lysate volume, as necessary. The extracted RNA was treated using the DNA-free DNA Removal kit (Invitrogen, Carlsbad, CA, USA) prior to reverse transcription using the SuperScript III First-Strand Synthesis kit for Real-Time PCR (Invitrogen), according to the manufacturer’s instructions. The DNA was quantified using a Qubit 3.0 Fluorometer (Invitrogen, CA, USA). Primer Both cDNA and DNA were then used to prepare sequencing libraries targeting the V4 region of the 16S rRNA genes, using multiplex identifier tagged primers 519F—5′-CAGCMGCCGCGGTAA-3′ and 806RB—5′-GGACTACNVGGGTWTCTAAT-3′ [27,28]. The 16S rRNA genes were amplified using a two-step nested protocol as described by Wilson et al. [26]. In the first step, PCR amplification was performed in triplicates. The annealing temperature for the PCR reaction was set to 55C and 25 cycles. Multiplex identifier tagged amplicons were pooled in equimolar amounts for library construction. Libraries were sent to the Norwegian Sequencing Center (Oslo, Norway) for sequencing on a MiSeq platform using the MiSeq Reagent Kit v2 (Illumina, San Diego, CA, USA) with paired-end reads of 2 × 250 bp lengths.

### 2.4. Sequence Analyses

The retrieved sequences were stored in the European Nucleotide Archive (PRJEB47256) and processed using DADA2 in R [29]. Briefly, primers were removed, and sequences were quality checked based on the illumine quality score before using a quality-based static trim. Amplicon sequence variants (ASVs) were generated from dereplicated reads. After merging the complementary reads and chimera removal, the taxonomy of the ASVs was assigned using a trained Silva database, based on the Silva release SSU Ref NR v138 [30]. ASVs of mitochondria, chloroplasts, and Eukaryotes were removed. An approximate Maximum Likelihood tree was calculated using FastTree2 [31] based on alignments using mafft aligner [32]. NMDS analyses were conducted using weighted Unifrac distances, as well as Constrained Analysis of Principal Coordinates. A Venn diagram for the different seasons was constructed, and Indicator species analyses were employed for the different environmental data. The analyses were done using R 4.1.1. in Rstudio 1.4.1717 and the “vegan 2.5-7”, “tidyverse 1.3.1”, “phyloseq 1.36.0”, “indicspecies 1.7.9”, “ggplot2 3.3.5”, “RColorBrewer 1.1-2”, “forcats 0.5.1”, “ggpubr 0.4.0”, “patchwork 1.1.1”, “scales 1.1.1”, “VennDiagram 1.7.0”, and “ape 5.5” packages [33,34,35,36,37,38,39,40,41,42,43,44,45]. The metabolic potential of the different ASVs was inferred based on their location in a phylogenetic tree of fully sequenced organisms and the genomic assets of the closest relative in this tree using PiCRUST2 2.4.0 with standard parameters in bioconda 0.17.0 [46,47]. Based on enzyme commission (EC) numbers, carbohydrate-active enzymes (CAZymes) belonging to the classes of glycoside hydrolases (GH), polysaccharide lyases (PL), carbohydrate esterases (CE), carbohydrate-binding modules (CBM), and glycosyl transferase (GT), as well as marker genes, namely *nifH*, *nirS/nirK*, *norB*, *nosZ*, *narB, napA, hzsA*, *amoA-pmoA*, *hao, *nrfA**, *aprA*, *dsrA*, *cysH*, *soxB*, *mmoX*, *mcrA*, *psaA*, and *psbA,* were used for metabolic analyses (Appendix A). CAZymes classified as GH and CBM were only counted as GH. 

## 3. Results

### 3.1. Cell Numbers and Environmental Data

As reported previously, a sea-ice algae bloom occurred in the bottom part of the ice, with cell numbers of 1.9 × 10^5^ cells mL^−1^ on 21 May and 2.1 × 10^6^ ± 1.3 × 10^4^ cells mL^−1^ in June, dominated by the pennate diatom *Nitzschia frigida/neofrigida* [11]. This was confirmed by increasing bulk chlorophyll (Chl) *a* concentration in the lower part of the ice cores from <0.05 mg m^−3^ in winter to 6.25 mg m^−3^ in May, 3.15 mg m^−3^ on 4 June, and 1.04 mg m^−3^ on 17 June (Figure 2). Bulk sea-ice nitrate concentrations increased from 0.47 µM in January to 1.94 µM in April and decreased during spring to 0.92 µM on 17 June (Figure 2). The combined bacterial and archaeal cell numbers decreased from 3.5 × 10^4^ cells mL^−1^ in January to 6.4 × 10^3^ cells mL^−1^ on 21 May, before increasing again to 3.0 × 10^4^ cells mL^−1^ and 2.1 × 10^4^ cells mL^−1^ on 4 and 17 June (Figure 2). 

### 3.2. Bacterial and Archaeal Community Composition

The bacterial and archaeal community composition during the N-ICE2015 expedition was investigated using 16S rRNA gene amplicon sequences for community profiling analyses, henceforth shortened to “rel. abundance”, including standard deviations calculated over all samples. This relative abundance was used to calculate the cell abundance in combination with flow-cytometry-derived bacterial and archaeal cell counts. 

The most prominent change in the bacterial and archaeal communities is the shift from *Cand*. Nitrosopumilus dominating winter samples to the dominance of *Nitrincolaceae* in spring and *Polaribacter* in early summer. The bacterial community consisted of mainly *Gammaproteobacteria* (36.9 ± 6.3% rel. abundance), *Bacteroidia* (20.6 ± 12.9% rel. abundance), and *Alphaproteobacteria* (16.7 ± 3.2% rel. abundance), complemented with *Verrucomicrobiae* (3.2 ± 3.9% rel. abundance), members of the SAR324 clade (0.9 ± 0.9% rel. abundance), and several other classes with <0.5% average rel. abundance (Figure 3). The archaeal community consisted of four classes, with *Nitrososphaeria* (16.4 ± 16.1% rel. abundance; Figure 3) being the most abundant and *Halobacteria*, *Nanoarchaeia*, and *Thermoplasmata* being <0.5% rel. abundance. On this level, a distinct shift from a winter community with high rel. abundance of *Nitrososphaeria* (27.8 ± 10.6% rel. abundance) to a spring/ early summer community with high rel. abundance of *Bacteroidia* (33.6 ± 3.5% rel. abundance) was visible. *Gammaproteobacteria* showed higher rel. abundance in spring/early summer (41.2 ± 4.8% rel. abundance) as compared to winter (33.2 ± 4.7% rel. abundance), while *Alphaproteobacteria* remained constant in rel. abundance ~18.6% (Figure 3).

The *Nitrososphaeria*, dominated by *Cand*. Nitrosopumilus increased from initial 13.3% rel. abundance in January towards 37.9% of rel. abundance in March and decreased to 0.6% in late June (Figure 3). This is reflected by an increase in cell abundance from 4.7 × 10^5^ cells mL^−1^ in January to 7.9 × 10^3^ cells mL^−1^ in March, the highest abundant taxa of the whole dataset, before decreasing to <3.5 × 10^2^ cells mL^−1^ in spring and summer (Figure 4). This dominance of *Cand*. Nitrosopumilus was due to only two ASVs, contributing with 35.1% rel. abundance to the total bacterial and archaeal community on 12 March, thus being the most abundant ASVs in the winter samples. 

*Bacteroidia* decreased from 19.0% in January to 7.3% in March and increased to 35.7% in May, with high rel. abundance of 29.6% on 4 June and 35.5% on 17 June (Figure 3). The high rel. abundance in January was mainly due to *Lutibacter* and members of the NS9 Marine Group (Figure 3), which showed cell abundance of 1.5 × 10^3^ and 1.1 × 10^3^ cells mL^−1^ (Figure 4). In May, the most dominant groups were *Ulvibacter* and *Saprospiraceae*, followed by *Lutibacter*, *Polaribacter*, *Chitinophagales*, *Lewinella*, *Vicingus*, and members of the NS9 and NS11-12 Marine Groups (Figure 3). During the sea-ice algae bloom *Ulvibacter*, members of the NS9 clade, and *Polaribacter* increased in abundance (Figure 3 and Figure 4). On 17 June, *Polaribacter* increased to 2.9 × 10^3^ cells mL^−1^, and *Lewinella* increased to 1.5 × 10^3^ cells mL^−1^, which constituted 13.9% and 7.2% rel. abundance, while *Ulvibacter*, *Lutibacter*, *Saprospiraceae*, and *Chitinophagales* decreased simultaneously (Figure 3 and Figure 4). On the level of ASVs, this succession was dominated by single ASVs for *Lewinella*, *Saprospiraceae*, *Chitinophagales*, *Vicingus*, and members of the NS11-12 Marine Group, while *Lutibacter*, *Polaribacter*, members of the NS9 Marine Group, and *Ulvibacter* were dominated by two ASVs, and only within the *Cryomorphaceae*, six ASVs contributed distinctively to the rel. abundance. 

*Gammaproteobacteria* decreased from 39.2% in January to 28.1% rel. abundance in March and increased to 42.1% in May and to 46.3% on 4 June (Figure 3). Within the *Gammaproteobacteria*, the initially high rel. abundance is based on eleven groups, with *Colwellia, Kangiellaceae*, and *Nitrincolaceae* being the most dominant, reaching 2.7 × 10^3^ cells mL^−1^, 2.2 × 10^3^ cells mL^−1^, and 2.0 × 10^3^ cells mL^−1^ (Figure 3 and Figure 4). This was reduced to only five groups in March, where *Nitrincolaceae* was dominant with 11.6 ± 2.0% rel. abundance (Figure 3), increasing to 2.4 × 10^3^ cells mL^−1^ and 2.3 × 10^3^ cells mL^−1^ in this month (Figure 4). The transition from winter to spring was marked mainly by an increase in *Nitrincolaceae* to 19.1% rel. abundance on 4 June and *Kangiellaceae* to 9.2% rel. abundance on 17 June (Figure 3). The peak of *Nitrincolaceae* on 4 June marked the highest cell abundance of a single taxon during the ice algae bloom with 5.7 × 10^3^ cells mL^−1^ (Figure 4). This succession of *Gammaproteobacteria* was driven mainly by two ASVs assigned to *Nitrincolaceae*, of which one was dominant in winter and May, while the other ASV was abundant in the June samples. Members of the OM43 and OM60 clades, the SUP05 cluster, *Kangiellaceae*, *Arenicella*, and *Paraglaciecola*, were dominated by a single ASVs, while members of the SAR92 and OM182 clades were dominated by two ASVs. Only members of the SAR86 clade with four simultaneously occurring ASVs, and *Colwellia* with one dominant ASV in winter, a second dominant ASV in May, and a third in June showed higher diversity on ASV level.

*Alphaproteobacteria* were stable at 18.6 ± 1% rel. abundance throughout the sampling period with decreased rel. abundance of 11.6% and 12.6% on 21 May and 17 June. This was mainly due to a constant abundance of one ASV of the SAR11 clade Ia, which decreased from 3.3 × 10^3^ cells mL^−1^ in January to 2.6 × 10^2^ cells mL^−1^ in May, only to increase again to 3.2 × 10^3^ cells mL^−1^ on 4 June (Figure 4). In contrast, the SAR11 clade II and the SAR11 clade decreased on 21 May and 17 June (Figure 3 and Figure 4). The three main ASVs of *Octadecabacter* increased in rel. abundance on 22 April and 21 May, just to decrease again in summer, while a single ASV of *Sulfitobacter* increased in rel. abundance in May and June (Figure 3); however, the cell abundances of both were constantly below 5.0 × 10^2^ cells mL^−1^ with the exception of Sulfitobacter on 17 June (Figure 4).

*Verrucomicrobiae* were found with 1.2 ± 0.5% rel. abundance in the winter samples but increased to 11.6% rel. abundance on 17 June (Figure 3), where they were the second most abundant taxon with 2.5 × 10^3^ cells l^−1^, second only to Polaribacter (Figure 4). This increase was due to a single ASV assigned to *Lentimonas* becoming the most dominant ASV in the community on that day. On the contrary, the members of the SAR324 clade peaked in April, just to decline to <0.1% rel. abundance in June (Figure 3). *Bdellovibrionia* and *Desulfuromonadia* both showed peaks in January with 1.1% and 2.2% rel. abundance.

### 3.3. Seasonal Variation

Constrained Analysis of Principal Coordinates based on the rel. abundance using the variables with the highest explanatory power (season, ice algae, and days) showed differences between the samples, confirming the differences of the communities between seasons and in correlation to the sea-ice algae bloom (Figure 5). The ice thickness remained fairly constant throughout N-ICE 2015, ranging from 92 cm to 138.5 cm until the floes reached the marginal ice zone, where bottom melting occurred (Appendix A; [48]).

In winter, 736 ASVs were found (293 unique ASVs), as compared to 173 ASVs (15 unique ASVs) in spring and 351 ASVs (25 unique ASVs) in summer, with 121 ASVs being present in all communities (Figure 6). Using Indicator species analyses, 51 indicative ASVs were found for winter, as compared to eight ASVs for summer. Most of these ASVs were found in low rel. abundance and are thus difficult to interpret given the potential methodological errors and the uncertainties of the importance of low abundance ASVs. However, one ASV of the SAR11 clade was found significantly higher in rel. abundance in winter samples (*p*-value = 0.012), whereas the main ASV of *Nitrincolaceae* showed significantly higher rel. abundance in summer with 8.2 ± 0.3% (*p*-value = 0.012). In addition, one ASV assigned to *Sulfitobacter* was significantly higher in spring and summer with >1% rel. abundance (*p*-value = 0.042). 

### 3.4. Bacterial and Archaeal Activity Estimations

Using the relative sequence abundance of 16S rRNA copies as a proxy, the activity of the bacterial and archaeal community was estimated and is stated as “rel. cDNA abundance”. Overall, the rel. cDNA abundance and the rel. abundance are congruent with the most abundant taxa also being the most active (Figure 3). The major finding is the decrease in rel. cDNA abundance of *Cand*. Nitrosopumilus from winter to spring, when the overall active *Kangiellaceae* and *Nitrincolaceae* dominated, and early summer, when *Polaribacter*, *Lewinella,* and *Lentimonas* were most active. The overall activity of the bacterial community was dominated by *Gammaproteobacteria* (63.3 ± 7.0% rel. cDNA abundance), *Bacteroidia* (12.0 ± 6.8% rel. cDNA abundance), and *Alphaproteobacteria* (8.9 ± 2.5% rel. cDNA abundance), complemented by *Verrucomicrobiae* (1.7 ± 2.8% rel. cDNA abundance), *Nitrospinia* (1.2 ± 1.2% rel. cDNA abundance), *Desulforomonadia* (0.8 ± 1.5% rel. cDNA abundance), *Dehalococcoida* (0.7 ± 0.8% rel. cDNA abundance), *Bdellovibrionia* (0.7 ± 0.7% rel. cDNA abundance), and members of the SAR324 clade (0.5 ± 0.4% rel. cDNA abundance; Figure 3). Within the Archaea *Nitrososphaeria* the rel. cDNA abundance of the dominant *Cand*. Nitrosopumilus increased from 8.7% in January to 19.6 ± 2.5% in March and decreased to <0.1% during spring and early summer (Figure 3). 

*Bacteroidia* decreased from 13.3% rel. cDNA abundance in January to 5.8 ± 0.9% during March and April and increased to 19.1% in May. On 4 June, the rel. cDNA abundance was decreased again to 12.2% before finally increasing to 22.4% on 17 June. The activity patterns in winter showed a high similarity to the abundance patterns of *Polaribacter*, members of the NS9 clade, *Lutibacter*, *Ulvibacter*, *Lewinella*, *Cryomorphaceae*, and *Saprospiraceae* (Figure 3). During the sea-ice algae bloom, *Lewinella*, *Polaribacter*, *Cryomorphyceae*, and members of the NS11-12 Marine Group increased in rel. cDNA abundance on 17 June, while *Vicingus* and *Saprospiraceae* decreased to <0.5% rel. cDNA abundance (Figure 3). 

*Gammaproteobacteria* rel. cDNA abundance was stable in winter with 59.7 ± 4.8% and peaked at 73.3% and 71.2% rel. cDNA abundance in May and on 4 June (Figure 3). *Kangiellaceae* and *Nitrincolaceae* were dominant in rel. cDNA abundance throughout the dataset, complemented by *Paraglaciecola*, *Colwellia*, *Arenicella*, *Candidatus* Endobugula, *Saccharospirillaceae,* and members of the OM182, SAR92 clades, and the SUP05 cluster. In winter, the rel. cDNA abundance of *Kangiellaceae* decreased from January to March, while the rel. cDNA abundance of *Nitrincolaceae* increased simultaneously (Figure 3). In the spring and early summer, the rel. cDNA abundance of *Nitrincolaceae* decreased to 11.7% on 17 June, while *Kangiellaceae* increased to 26.7% (Figure 3). *Colwellia* peaked in rel. cDNA abundance on 5 March, while the members of the OM182 clade peaked in April and May (Figure 3). The rel. cDNA abundance of members of the SUPt05 cluster was highest on 12 March and 4 June, *Arenicella* was highest in May, similar to *Candidatus* Endobugula, while members of the SAR92 were highest on 4 June, and *Paraglaciecola* peaked on 21 May and 17 June, showing a succession among the *Gammaproteobacteria* (Figure 3).

The rel. cDNA abundance of *Alphaproteobacteria* was stable with 9.1 ± 0.1% in winter, increased to 13.3% in spring/ early, and decreased again to 8.1% on 17 June (Figure 3). SAR11 Clade Ia peaked in rel. cDNA abundance on 4 June (Figure 3). *Sulfitobacter* and *Octadecabacter* were low in winter but increased in rel. cDNA abundance on 17 June (Figure 3). 

The rel. cDNA abundance of *Verrucomicrobiae* was stable at 0.7 ± 0.2% but increased on 17 June to 8.1% due to an increase in *Lentimonas* rel. cDNA abundance (Figure 3). Members of the SAR324 clade showed a rel. cDNA abundance <1% (Figure 3). Congruent with the peaks in rel. abundance in January, *Bdellovibrionia* and *Desulforomonadia* showed peaks in rel. cDNA abundance of 2.2% and 4.1% (Figure 3). 

### 3.5. Functional Approximation Based on PiCRUST2

The metabolic potential of the different ASVs inferred using PiCRUST2 bears problems in the form of representativeness of the inferred genome. This is especially true at ASV level and in underrepresented environments, which are both not covered well in the databases. Therefore, even the genomes determined as closest to the ASV might still differ to some extent from the true genome of the ASV [49]. Therefore, the results of these analyses have to be treated carefully and represent only an estimation, not measured metagenomes. The predictions using PiCRUST2 resulted in 827 “metagenomes” with weighted Nearest Sequenced Taxon Index values < 1.0. NSTI indicates the phylogenetic distance to the nearest neighbor in the genome database. Among the CAZymes, all classes besides PLs were found with >0.2% rel. abundance and showed an increase from winter to spring and early summer (Figure 7). GHs increased significantly (*p* = 0.011; *t*-test) from 0.41 ± 0.11% to 0.95% in spring and 0.80 ± 0.18% in early summer (Figure 7). GTs were stable at 0.61 ± 0.03% rel. abundance, while CE and CBM increased significantly (*p* = 0.006 and 0.023; *t*-test) from 0.17 ± 0.03% and 0.23 ± 0.03% to 0.29% and 0.34% in spring and 0.24 ± 0.01% and 0.31 ± 0.05% in early summer (Figure 7). The distribution of CAZymes was relatively even among the Gammaproteobacteria, with GH and PL mostly found in Glaciecola, Paraglaciecola, and Colwellia; CE in Cellvibrionaceae; and CBM and GT in Colwellia, UBA10353 Marine Group, and Pseudomonas. In addition, members of the SAR86 and OM182 clades showed a high abundance of GT. Within the Bacteroidia, GH, CE, CBM, and GT distribution was dominated by Saprospiraceae, Cryomorphaceae, and members of the NS9 Marine Group, while PL were found almost exclusively in Polaribacter. Alphaproteobacteria possessed mainly GT and some CBM, dominated by members of the SAR11 Clade II and Magnetospiraceae. The CE found in Verrucomicrobiae were mainly predicted for Roseibacillus and Rubritalea. 

Among the marker genes, only *nirS/nirK* (Denitrification), *amoA* (Nitrification), *aprA* (Dissimilatory sulphate reduction), and *cysH* (Assimilatory sulfate reduction) reached a rel. abundance of >0.05%, while *nifH* (Nitrogen fixation), *norB* (Denitrification), *nosZ* (Denitrification), *narB* (Assimilatory nitrate reduction), *hao* (Nitrification), *nrfA* (Dissimilatory nitrate reduction to ammonium), and *dsrA* (Dissimilatory sulfite reduction) were predicted in low abundance (<0.01%), and *hzsA* (Annamox), *soxB* (Sulphur oxidation), *mmoX* (Aerobic methane oxidation), *mcrA* (Methanogenesis), *psaA* (Photosynthesis), and *psbA* (Photosynthesis) were not predicted (Figure 7). Since the *amoA* and *pmoA* genes are very similar and the genes were almost exclusively predicted for *Cand*. Nitrosopumilus, known to possess *amoA* [50], the predictions were considered as *amoA*. The *amoA*, as well as *nirS/nirK* were most abundant in winter samples with 0.10 ± 0.05% and 0.05 ± 0.02%, rel. abundance and decreased significantly (both *p* = 0.011) to <0.01% in spring and early summer (Figure 7). The abundance of *amoA* was clearly dominated by *Cand*. Nitrosopumilus, but also predicted for *Gammaproteobacteria* (Figure 7). Within the *Bacteroidia*, *nirS/nirK* and *norB* were predicted for *Aurantivirga*, *Flavobacterium*, and *Tenacibaculum*. Most genes for denitrification predicted for *Gammaproteobacteria* were found in *Colwellia* and *Pseudomonas* (*nirS/nirK* and *nosZ*, Figure 7). For the archaeal *Nitrosphaeria* only the abundance of *nirS*/*nirK* was predicted, distributed mainly to *Cand*. Nitrosopelagicus and *Nitrosopumilaceae*. The marker gene for assimilatory nitrate reduction (*narB*) was found in *Polaribacter*, *Arcobacteraceae*, and *Synechococcus*, while nitrogen fixation (*nifH*) was predicted for *Bdellovibrionia* (members of the OM27 clade), *Campylobacteria* (*Arcobacteraceae*), and *Desulfuromonadia* (members of the PB19 clade; Figure 7). 

The *aprA* gene was present in winter with 0.05 ± 0.01%, decreased significantly (*p* = 0.032) to 0.01% in spring and increased again to 0.03 ± 0.01% in early summer. The marker genes for dissimilatory sulfate reduction and oxidation (*aprA* and *dsrA*) were predicted for members of the OM27 clade (*Bdellovibrionia*), *Desulfofrigus* (*Desulfobacteria*), *Desulforhopalus* (*Desulfobulbia*), and PB19 (*Desulfuromonadia*), and members of the SAR11 clades Ia, II, and further unspecified members, as well as members of the AEGEAN-169 Marine Group only *aprA*. The *cysH* gene was predicted stable at 0.11 ± 0.01% rel. abundance in all samples and for a large variety of classes, dominated by *Cryomorphaceae* and members of the NS9 Marine Group in the *Bacteroidia*, while *Magnetospiraceae* dominated the abundance in the *Alphaproteobacteria,* members of the OM182 clade the *Gammaproteobacteria*, and *Lentimonas* and *Roseibacillus* the *Verrucomicrobia*.

## 4. Discussion

Here, we describe the transition of the bacterial and archaeal communities of Arctic sea ice from winter to spring and early summer. Even though this is not a true time series due to the necessary relocation in drifting ice, the samples taken on Floe 3 (April to June) show a clear succession of the bacterial and archaeal community. The community from April matches the communities from January and March (Floes 1 and 2), while the community from 17 June (Floe 4) is similar to 4 June (Floe 3). Thus, suggesting comparability of samples. In addition, the high taxonomic resolution of the ASVs supports the comparability since most ASVs were found in most or all samples. 

An ice algal bloom of the pennate diatom *Nitzschia frigida* dominated the bottom ice algal assemblage of both FYI and SYI in May and June. Other pennate diatoms, unidentified flagellates, and resting cells (particularly of the dinoflagellate *Polariella glacialis*) came next in importance, while centric diatoms and ciliates occurred at low abundances (Appendix A; [11]). *Melosira arctica* occurred at low abundances, and the only records of this species were from slurp gun samples taken by divers below sea-ice ridges. The characteristic mats of this species were not observed. The ice algal bloom corresponds to an increase in light availability in the ice, enabling phototrophic growth. A decrease in nitrate concentrations was only observed around 22 May, when sea-ice algae cell numbers increased more rapidly [11]. A succession of the bacterial and archaeal community based on the algae-derived carbon sources was observed in the bottom layer of the sea ice from winter to early summer.

### 4.1. The Winter Community

The most drastic change was the vanishing of *Cand*. Nitrosopumilus, from high abundance in winter to near absence in summer, as is congruent with the findings in Arctic surface waters [7,26,51,52]. *Cand*. Nitrosopumilus performs nitrification [17], thus explaining the high rel. abundance of *nirS/nirK* and *amoA* predicted for *Cand*. Nitrosopumilus. This indicates that ammonium oxidation to nitrite could be a key metabolism in winter months and, combined with nitrite oxidation, could lead to the replenishment of the nitrate pool in the sea ice [53]. The nitrite oxidizing *Nitrospina*, often associated with *Cand*. Nitrosopumilus [54,55,56], was found in low rel. abundance < 1% but high rel. cDNA abundance in winter samples (~3–4% rel. cDNA abundance). However, gammaproteobacterial *Nitrincolaceae* (formerly *Oceanospirillaceae*) may also use nitrite oxidation as the main energy source, thus explaining the relatively high abundance during winter and adding to the production of nitrate [57,58]. Photoinhibition was assumed to be a factor for *Cand.* Nitrosopumilus mortality and would explain the near absence in light conditions in spring and early summer [10,59,60]. In addition, ammonium is the most easily accessible nitrogen source for algae, which leads to competition for ammonium when light is present [9,61]. Even though *Cand*. Nitrosopumilus is adapted to very low ammonia concentrations, the competition could add to the explanation of the decreasing rel. cDNA abundance of *Cand.* Nitrosopumilus in April and the near disappearance of this taxon during the sea-ice algae bloom. However, this would not explain the decreased rel. cDNA abundance in January, which might correlate to the lower abundance in this sample. In addition, competition with fast-growing members of the *Gammaproteobacteria* and *Bacteroidia* could lead to a decrease in *Cand.* Nitrosopumilus in carbon-replenished spring and summer samples [59,62,63]. 

Members of the alphaproteobacterial SAR11 clade are resilient, slow-growing organisms with low requirements for carbon sources and high abundance in oligotrophic polar oceans [64,65,66,67], thus explaining the indicator function of one ASV of the SAR11 Clade in winter. The abundance of the *aprA* gene in winter samples is due to the abundance of members of SAR11 clades. *AprA/B* has been found in SAR11 previously, and dissimilatory sulfate oxidation has been suspected [68,69]. The higher abundance of genes indicating nitrogen and sulfur metabolisms in winter samples supports the hypothesis of a bacterial and archaeal community, where limited carbon sources lead to shifts towards other, e.g., nitrogen-based metabolisms. In addition, members of the SUP05 cluster, reported in Antarctic surface waters, harbor the genetic requirements for chemolithoautotrophic carbon assimilation using nitrate and reduced sulfur compounds [52,58,70]. Members of the SAR324 clade were found to use lithotrophy, heterotrophy, and alkane oxidation simultaneously [71] and have been found in Arctic surface and under-ice waters [7]. Still, their role in this ecosystem remains unknown. The abundance of sulfur-reducing *Desulfuromonadia* [72] also points toward alternative metabolisms in winter. 

### 4.2. A Carbon-Based Succession in Spring/Early Summer

*Gammaproteobacteria* and *Bacteroidia* are usually found in eutrophic, carbon-rich marine environments, often in correlation with phytoplankton blooms, thus explaining their abundance in the spring and early summer samples [8,73,74,75,76,77,78,79]. The most abundant Gammaproteobacteria in these samples, *Nitrincolaceae*, members of the OM182 and SAR92 clades, and the *Alteromonadaceae* groups *Arenicella*, *Paraglaciecola*, and *Colwellia*, have been found in association with phytoplankton blooms and polar waters [8,80,81,82,83]. *Nitrincolaceae* harbor a high diversity of carbon metabolisms, explaining their abundance during the sea-ice algae bloom [83,84]. *Kangiellaceae* were suggested to extracellularly utilize amino acid as carbon and nitrogen sources, explaining the abundance of *Kangiellaceae* in early winter and summer samples [85]. *Bacteroidia* harbor a multitude of carbon degradation pathways, leading to successions of different groups after phytoplankton blooms [8,77,79]. In various marine and sea-ice environments, *Polaribacter* co-occur with phytoplankton blooms [8,19,67,86,87], explaining the high abundance of this group in spring and early summer. High abundances of members of the NS9 clade have also been found in summer and fall in samples from Antarctica and were correlated to phytoplankton appearance in early spring in the Southern Ocean [88,89]. In addition, members of the *Verrucomicrobiae*, specifically *Lentimonas,* degrade complex carbon sources, and hence have been found in later stages of phytoplankton blooms [8,90,91,92]. *Rhodobacteraceae*, such as *Sulfitobacter* and *Octadecabacter,* are often correlated to diatom blooms [8,83,93]. Thus, a sea-ice algae-derived carbon-source-driven succession of the bacterial and archaeal community in spring and early summer is congruent with the finding of significantly different communities between seasons and with ASVs of *Nitrincolaceae* and *Sulfitobacter* being indicator species for spring and early summer. This succession might even go on after early summer, as samples from Arctic sea ice and brines with a high abundance of different taxa within the *Gammaproteobacteria* and *Bacteroidia* indicate [16,21,22]. One explanation for these differences could be that a further succession throughout summer and fall could lead to increased carbon source production, which then could fuel even the winter community. This may explain the higher abundance of several taxa within the *Gammaproteobacteria* (*Nitrincolaceae*, *Kangiellaceae*, and *Colwellia*) and *Bacteroidia* (*Lewinella*, *Lutibacter*, *Polaribacter*, and members of the NS9 marine group). However, due to the lack of fall samples in this study and the geographical distances between the studies, this has to remain a hypothesis.

### 4.3. CAZyme Abundance Support Succession Based on Carbon Sources

The pattern of increased predicted CAZyme abundance in Bacteroidia and Gammaproteobacteria in spring and early summer, as compared to winter, is based on the abundance of these taxa. Therefore, an increase in CAZymes indicates the potential of the various genera within the Bacteroidia and Gammaproteobacteria to utilize different carbon sources and, consequently, different ecological niches. These community patterns are also congruent with the finding in a sample from adjacent first-year ice (FYI), where a similar bloom occurred [11], and the pelagic communities in the *P. pouchetii* bloom observed below the ice starting on 25 May [6], where a succession of the bacterial and archaeal community in the surface waters from *Alphaproteobacteria* and *Cand*. Nitrosopumilus in winter to *Gammaproteobacteria* and *Bacteroidia* in spring and early summer was found [7]. This correlation seems to be independent of the bloom-forming species, implying that the variety of carbon sources provided by the bloom-forming algae is broadly similar. This implies that functional guilts in the bacterial and archaeal community are not specialized to a single algal species, even though specific ASVs might be dependent on specific algae species [93]. This could also explain the similarity to the algae-derived carbon-source-based bacterial and archaeal successions found in phytoplankton bloom in the arctic as well as temperate waters [7,8,77].

## 5. Conclusions

Here, we report the community composition and activity of Arctic sea-ice *Bacteria* and *Archaea* during the transition from winter to early summer in 2015 north of Svalbard. We show the transition of this community from a winter state with decreasing cell numbers, mostly nitrogen-based metabolisms, and dominance of *Cand*. Nitrosopumilus from January to an early spring state in March. There, nitrification potentially replenished the nitrate pool, which could then be used by other organisms during spring and summer. In spring, *Bacteroidia* and *Gammaproteobacteria* increased in abundance, while *Cand*. Nitrosopumilus nearly vanished. The succession of *Gammaproteobacteria* and *Bacteroidia* is driven by season and the corresponding availability of carbon sources derived from the light-induced, diatom-dominated ice algal spring bloom, which leads to the establishment of different ecological niches, subsequently exploited by the best-adapted groups. Given the changes in arctic environments based on global warming, this seasonal succession from dominantly nitrogen-based to dominantly carbon-based cycling could be interrupted or accelerated by alteration in the sea-ice environment, potentially leading to changes in the microbial sea-ice ecosystem.

## Figures and Tables

**Figure 1 microorganisms-10-01618-f001:**
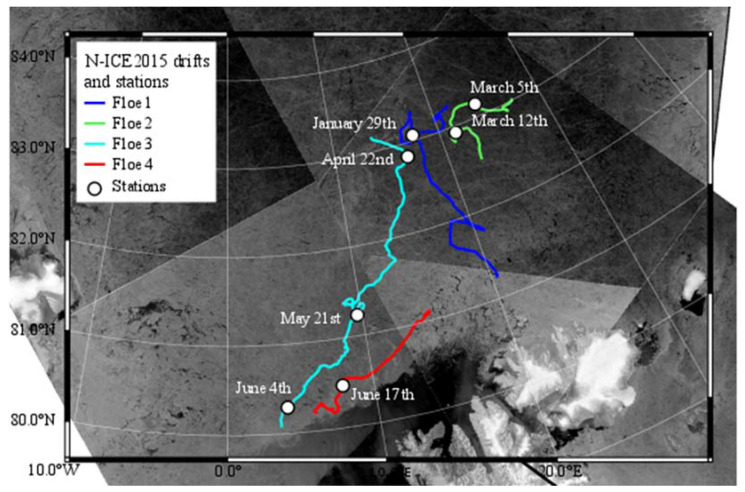
Map of the sampling area with the four different ice Floe drift tracks color-coded and the ice core samples used in/available for this study indicated by date. Image source: RADARSAT-2 image provided by NSC/KSAT under the Norwegian-Canadian RADARSAT agreement. RADARSAT-2 Data and Products^©^ MacDonald, Dettwiler and Associates Ltd. (2013) All Rights Reserved. RADARSAT is an official mark of the Canadian Space Agency. Map created by the Norwegian Polar Institute.

**Figure 2 microorganisms-10-01618-f002:**
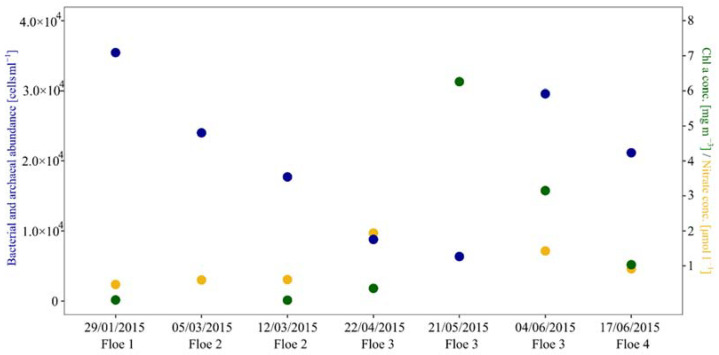
Bacterial and archaeal cell abundance (blue), chlorophyll *a* concentration (green), and nitrate concentration (yellow) during the five months of the N-ICE2015 observation. Gaps in the graph are due to missing values on these days.

**Figure 3 microorganisms-10-01618-f003:**
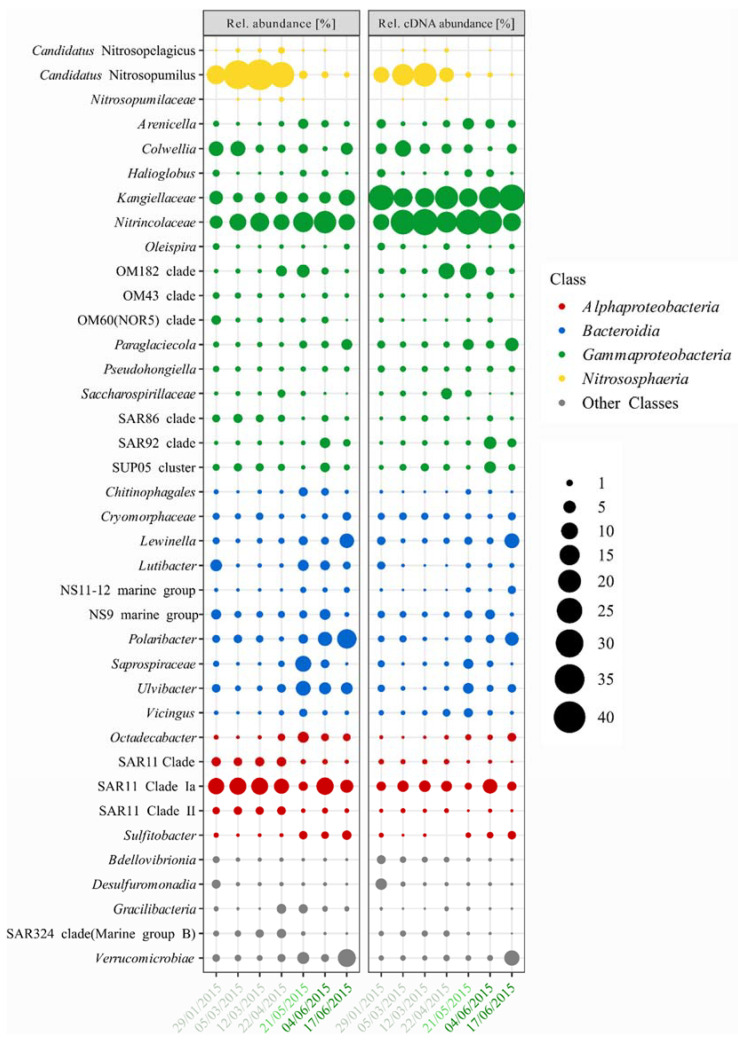
Bubble plot of the most abundant bacterial and archaeal taxa during the N-ICE2015 expedition, which constitute at least 80% of the total community per sample. The plot shows the distribution of 16S rRNA genes (rel. abundance) and RNA the distribution of 16S rRNA copies (rel. cDNA abundance) of the different taxonomic groups. The color of the dates indicates the season (light green = winter, green = spring, and dark green = summer) and the ice algae bloom (dark green).

**Figure 4 microorganisms-10-01618-f004:**
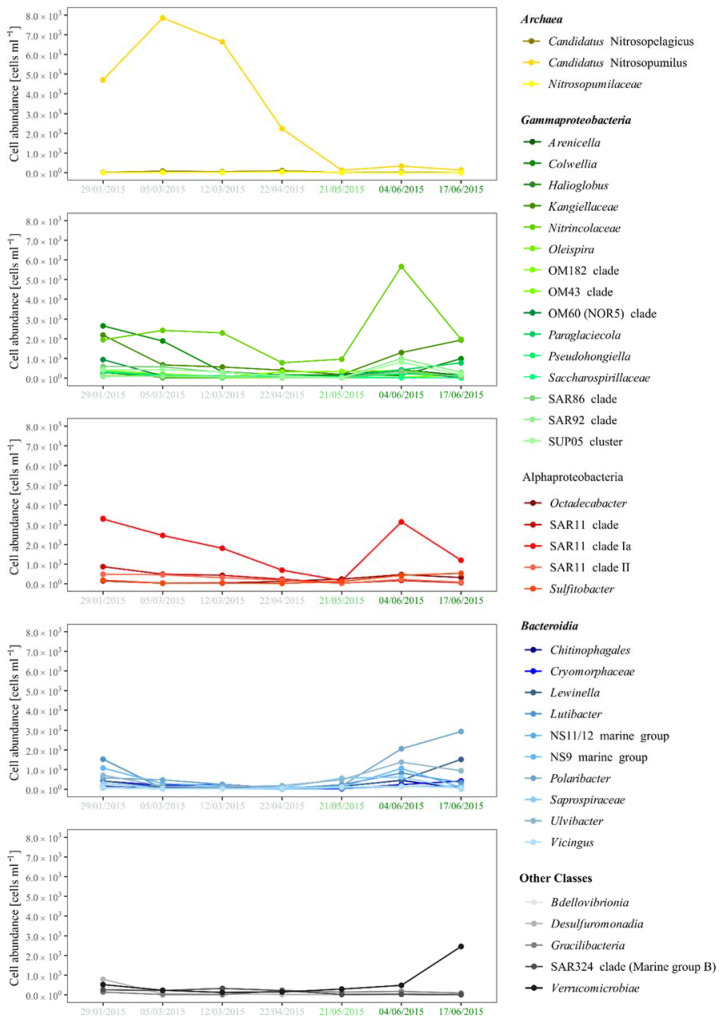
Line plots of bacterial and archaeal abundance of the most abundant taxa based on the rel. abundance and the flow cytometry derived cell numbers. The color of the dates indicates the season (light green = winter, green = spring, and dark green = summer) and the ice algae bloom (dark green).

**Figure 5 microorganisms-10-01618-f005:**
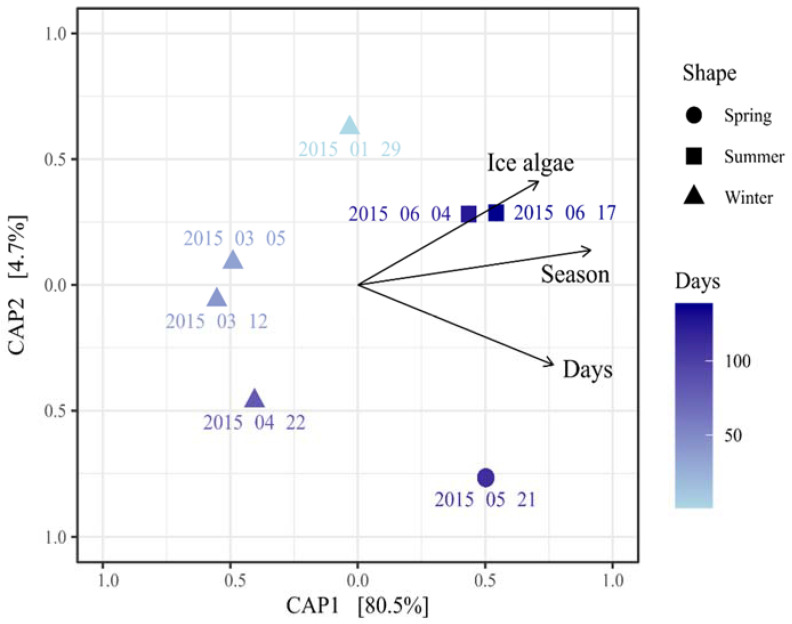
Constrained analysis of principal coordinates analyses including season, ice algae, and days as main factors determining the bacterial and archaeal community.

**Figure 6 microorganisms-10-01618-f006:**
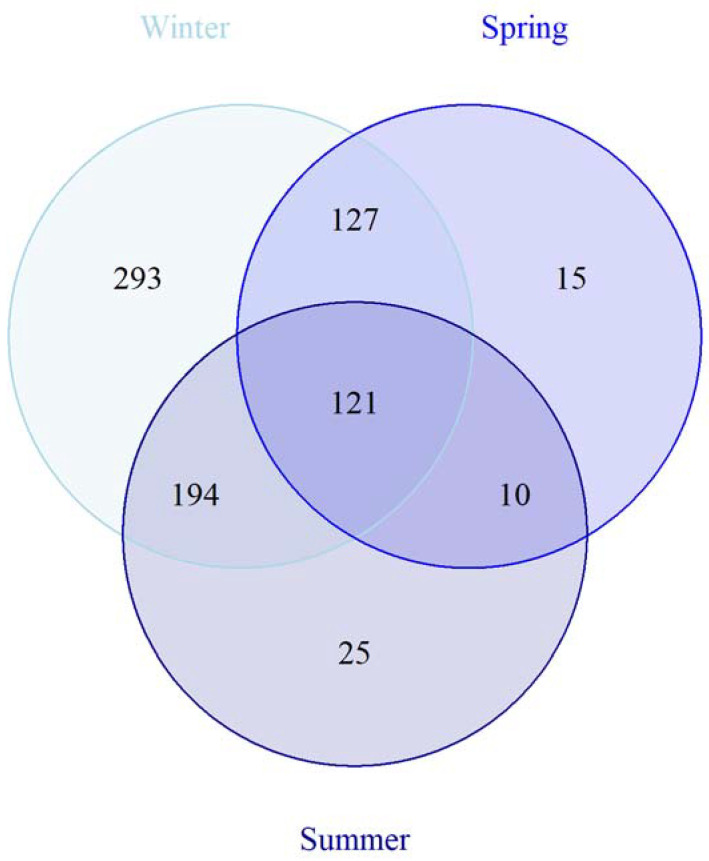
Venn diagram comparing the abundance of the different ASVs for each season (winter = light blue, spring = blue, and summer = dark blue). Each circle represents a season with the number of ASVs specific for the season stated inside. Overlapping areas show the occurrence of specific ASVs in two or all three seasons.

**Figure 7 microorganisms-10-01618-f007:**
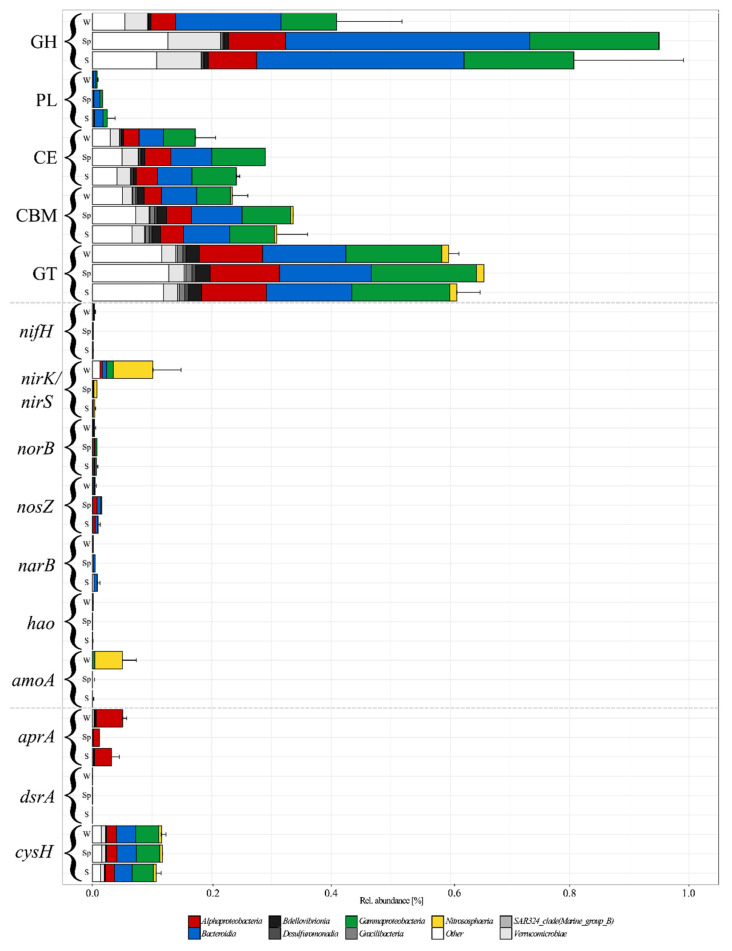
Rel. abundance of predicted CAZyme classes (glycoside hydrolases (GH), polysaccharide lyases (PL), carbohydrate esterases (CE), carbohydrate-binding modules (CBM), and glycosyl transferase (GT)) and marker gene abundances for the three distinct seasons (Winter (W), Spring (Sp), and Summer (S)). The rel. abundance is shown as % of total genes detected in the PiCRUST2 inferred metagenome for each sample. The error bars show the variability in the four stations pooled for the winter sample and the two stations for the summer sample. No error bar is available for spring since only one sample was taken in this season. The bars include the predicted taxonomic assignment of the respective CAZyme classes and marker genes on class level.

## Data Availability

All sequences are stored in the European Nucleotide Archive (PRJEB47256).

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
