# Peer review of "A Winter-to-Summer Transition of Bacterial and Archaeal Communities in Arctic Sea Ice"

_microorganisms, 2022, doi:10.3390/microorganisms10081618_

Round 1
Reviewer 1 Report
The manuscript by Thiele and colleagues describes changes in the microbiological community of Artic sea ice dependent on the season. Different samples were analyzed by means of Bacterial counts as well as NGS techniques (concerning both DNA & RNA). Overall, the manuscript is well written, the results are clearly presented and the study is, in my opinion, of high value for the scientific community.
There is not much to comment from my side, only a few minor corrections are suggested.
Abstract, line 27: Nitrosopumilus „vanished“ is a bit bold, maybe you could say that it was not detected.
Figure 2: Please say “Bacterial and archaeal cell abundance” to be in line with the graph
Line 234: “Kangiellaceae” should be italic
Line 333-335: The format is a bit weird, is that intentional to describe the drawback of the method or a mistake?
Author Response
Dear reviewer,
thank you very much for the constructive review. Your work is much appreciated. Please find the responses to the individual comments below and as track-changes in the re-submitted manuscript.
Abstract, line 27: Nitrosopumilus „vanished“ is a bit bold, maybe you could say that it was not detected.
- The sentence was changed accordingly.
Figure 2: Please say “Bacterial and archaeal cell abundance” to be in line with the graph
- The sentence was changed accordingly.
Line 234: “Kangiellaceae” should be italic
- The name was put into italics.
Line 333-335: The format is a bit weird, is that intentional to describe the drawback of the method or a mistake?
- The section has been reworked, hopefully explaining the problems of using PiCRUST2 on ASV and on environmental samples of little investigated environments.
Best regards
Stefan Thiele

Reviewer 2 Report
Dear Authors,
Is the manuscript clear, relevant for the field and presented in a well-structured manner?
I think that the study is valuable and relevant to the field. In general, the article is mostly clearly presented and the structure makes sense to me.
The comments on particular parts of the text are to be found below.
Specific comments:
In Introduction chapter, there is no references to similar studies conducted in Arctic sea. It would be clearer if the Authors concentrated on main results and removed the suggestion. Some sentences should be in the Results chapter instead.
Please, introduce more information about summer transition of bacterial and archaeal communities in Arctic sea ice.
Please, read the paper 1) by Nahui Olin Medina-Chávez & Michael Travisano (Archaeal communities: The microbial phylogenomic frontier, 2022), and 2) by Wei Qin et al. (Candidatus Nitrosopumilus, 2016).
Please, write the Introduction again.
Materials and methods
I am positively surprised by the very detailed description of the methods used.
How were algae (diatoms, Line 59: Nitzschia frigida, Line 72-73: Fragilariopsis cylindrica, Melosira arctica), bacterial and archaeal communities taxonomic verifications performed? Do the Authors have photographic documentation of species?
Line 534: water[7,8,73] should be water [7,8,73]
References
The following publications should be included in manuscript and References: 1) Schoenrock et al. 2015. Climate change confers a potential advantage to fleshy Antarctic crustose macroalgae over calcified species. Journal of Experimental Marine Biology and Ecology, 474:58-66; 2) Wiencke Christian (Ed.) 2011. Biology of Polar Benthic Algae. In the series Marine and Freshwater Botany.
Author Response
Dear reviewer,
thank you very much for the constructive review. Your work is much appreciated. Please find the responses to the individual comments below and as track-changes in the re-submitted manuscript.
Specific comments:
In Introduction chapter, there is no references to similar studies conducted in Arctic sea. It would be clearer if the Authors concentrated on main results and removed the suggestion. Some sentences should be in the Results chapter instead.
- We worked on the introduction, made respective changes, and added some references about Arctic sea-ice bacteria and archaea (references 12-16).
Please, introduce more information about summer transition of bacterial and archaeal communities in Arctic sea ice.
- This manuscript is to our knowledge the first study focussing on the transition of the bacterial and archaeal community from winter into summer in open ocean arctic sea ice.
Please, read the paper 1) by Nahui Olin Medina-Chávez & Michael Travisano (Archaeal communities: The microbial phylogenomic frontier, 2022), and 2) by Wei Qin et al. (Candidatus Nitrosopumilus, 2016).
- We have read both papers, however a paper about Archaea in general, their diversity, evolution, and systematics seems to us not to add much value to the introduction. Especially, since Nitrosopumilus is only mentioned in the review of Medina-Chaves & Travisano in terrestrial environments, but not in marine environments. Also there is no paragraph about marine or ice environments in this review. The chapter of Bergeys Manual is known to us, but we would argue that the paper about the discovery of Cand. Nitrosopumilus by Martin Könneke (2005) is a good reference for this candidate species. Therefore, we did not include the references into the introduction.
Please, write the Introduction again.
- We have modified the introduction. See comment above.
Materials and methods
I am positively surprised by the very detailed description of the methods used.
How were algae (diatoms, Line 59: Nitzschia frigida, Line 72-73: Fragilariopsis cylindrica, Melosira arctica), bacterial and archaeal communities taxonomic verifications performed? Do the Authors have photographic documentation of species?
- The description of Nitzschia frigida found in samples during the N-ICE expedition can be found in the reference 11 (Olsen et al., 2017). The other algae mentioned are only examples for algae found in Arctic waters or sea ice, but have not been found in this study. As we are focussing on the bacterial and archaeal community, we have no photographic documentation. However, pictures can be found in Olsen et. al. 2017. For the bacterial and archaeal community, such visual confirmation are impossible, given their very similar shape. Methods like Fluorescence in situ hybridization could have been used, but no samples were taken for such investigations during the expedition. Therefore, we unfortunately have no other taxonomic data as the sequences. We apologize for this.
Line 534: water[7,8,73] should be water [7,8,73]
- The space has been added.
References
The following publications should be included in manuscript and References: 1) Schoenrock et al. 2015. Climate change confers a potential advantage to fleshy Antarctic crustose macroalgae over calcified species. Journal of Experimental Marine Biology and Ecology, 474:58-66; 2) Wiencke Christian (Ed.) 2011. Biology of Polar Benthic Algae. In the series Marine and Freshwater Botany.
- We considered implementing the suggested references, however had to decide against it. We apologize for this, but this study is not focussing on ice-algae, but on bacteria and archaea, which react to the carbon provided by ice-algae. These ice-algae that provide the carbon sources are open ocean, pelagic microalgae, whereas the suggested reference studied mostly coastal, benthic, macroalgae. In sea-ice environments, these algae do not appear and hence have no connection to the bacterial and archaeal communities in sea-ice. Therefore, we decided against an implementation.
Best regards
Stefan Thiele
